# Reasons for declining to participate in a trial of online cognitive behavioural therapy following orthopaedic trauma: A mixed methods study

Sheila Sprague[1,2]*, Jodi L. Gallant[1], Natalie Fleming[1], Sofia Bzovsky[1], Sarah MacRae[1], Mavis Lyons[1], Jose Manuel De Maria Prieto[1], Herman Johal[1], Paula McKay[1], Jason W. Busse[2,3]*, on behalf of the COPE Investigators[¶]

1 Division of Orthopaedic Surgery, Department of Surgery, McMaster University, Hamilton, Ontario, Canada, 2 Department of Health Research Methods, Evidence, and Impact, McMaster University, Hamilton, Ontario, Canada, 3 Department of Anesthesia, Michael G. DeGroote School of Medicine, McMaster University, Hamilton, Ontario, Canada

¶ Membership of the COPE Investigators is provided in the Acknowledgments.
* sprags@mcmaster.ca (SS); bussejw@mcmaster.ca (JWB)

**Data Availability Statement:** Qualitative data cannot be shared publicly as it may contain

## Abstract

The timely enrollment of study participants is critical to the success of clinical trials. Understanding factors that contribute to patients' decision to participate in trials involving online cognitive behavioural therapy for pain management should prove helpful to optimize the design of study protocols. Fracture patients from an orthopaedic clinic who declined to participate in the Cognitive behavioural therapy to Optimize Post-operative rEcovery (COPE) trial were asked to complete a Research Participation Questionnaire that asked them about their previous experiences with clinical research and mental health therapy and their reasons for declining to participate in the COPE trial. At the end of the questionnaire, a subset of participants was offered the opportunity to participate in a telephone interview to further discuss why they declined to participate in the COPE trial. Sixty-four patients who declined to participate in the COPE trial completed the questionnaire and twenty of these participants agreed to take part in a telephone interview (31%). Twenty-two participants (34%) had previous experience with clinical research and six participants (9%) had received cognitive behavioural therapy (CBT) in the past. Excessive time commitment (41%) was the most commonly selected reason for not participating in the COPE trial, followed by a disinclination to participate in clinical research (19%). Four themes emerged from the interviews with participants: 1) belief that they could overcome mental health challenges after their fracture without external help; 2) belief that CBT might be helpful for some fracture patients, but not for themselves; 3) preference for online or in-person CBT; and 4) concerns regarding time commitment. To maximize enrollment, trials exploring the role of psychotherapy in recovery from orthopedic trauma should optimize time commitment of psychotherapy. Providing information in the patient consent process regarding evidence for psychotherapy and recovery from orthopedic trauma may also prove helpful in promoting patient enrollment.

identifiable information. All relevant data are within the paper.

**Funding:** The COPE trial is funded by the Canadian Institutes of Health Research (CIHR) (Project Grant: 156730, https://cihr-irsc.gc.ca), and the Orthopaedic Trauma Association (OTA) (https://ota.org). The Michael G. DeGroote Institute for Pain Research & Care at McMaster University funded start-up activities at the Methods Centre. JB is supported, in part, by a CIHR Canada Research Chair in Prevention & Management of Chronic Pain. The funding sources did not influence the design or conduct of the study; collection, management, analysis, and interpretation of the data; preparation, review, or approval of the manuscript; or decision to submit the manuscript for publication.

**Competing interests:** The authors have declared that no competing interests exist.

# Introduction

While clinical trials frequently struggle to enroll study participants in a timely manner [1], particularly in studies of surgical patients [2], trials evaluating cognitive behavioural therapy (CBT) face unique challenges. Participants must dedicate a substantial amount of time and effort towards therapy sessions, requiring a high level of participant engagement. Previous trials assessing the effectiveness of CBT found patients declined to participate in studies for a variety of reasons, including previous negative experiences with counselling and negative views towards psychotherapy [3].

Consequently, a randomized controlled trial of an online CBT program provides an opportunity to assess reasons for declining participation in a fracture clinic population. In this case, participants previously declined participation in the Cognitive behavioural therapy to Optimize Post-operative rEcovery (COPE) trial, which is a randomized controlled trial evaluating the effectiveness of online CBT on the reduction of chronic post-surgical pain (CPSP) in patients who have recently had surgery to repair an extremity fracture [4, 5]. The present study, completed alongside the COPE trial, addresses the questions: (1) "Why do orthopaedic trauma patients decline to participate in a trial involving online CBT for pain management?", and (2) "What beliefs do orthopaedic trauma patients who decline to participate in the COPE trial hold about CBT and clinical research?".

# Materials and methods

## Study design

This was a mixed methods study that consisted of: (1) a brief questionnaire exploring reasons why patients declined to participate in the COPE trial (quantitative phase), and (2) a telephone interview to further explore reasons for declining, as well as attitudes, knowledge of, and beliefs regarding clinical research and mental health interventions (qualitative phase). This study was conducted at one clinical site currently participating in the COPE trial (Hamilton Health Sciences–General Site) and followed the Consolidated criteria for reporting qualitative research (COREQ) criteria [6]. We obtained ethics approval for both the COPE trial and this study on December 1st, 2020 from the Hamilton Integrated Research Ethics Board (Project #11517) prior to the initiation of any study-related activities.

## COPE trial overview

Eligibility for the present study included having previously declined participation in the COPE trial. The COPE trial was designed to determine the effectiveness of an online CBT program with asynchronous therapist support, versus usual care, in reducing CPSP after fracture surgery. After providing informed consent, participants are randomized to receive either: (1) treatment as usual or (2) treatment as usual plus a 7-module, self-directed program of online CBT focused on pain management with asynchronous support from a therapist. CBT participants completed an initial telephone assessment with a therapist to confirm the participant's suitability for the program before being provided with access to the secure, online program. CBT program modules were designed specifically for the COPE trial and focus on the following components: emotional processing of the experience of pain and introduction to the cognitive behavioural mode; introduction to the biopsychosocial model of pain, cognitive strategies, behavioural strategies, mindfulness, and acceptance; and optimizing functioning and preparing for the future. Participants progress through modules that consist of self-assessment tools, educational materials, activities and homework. Participants in both study arms complete study questionnaires at 3, 6, 9 and 12 months after their fracture to assess the following

outcomes: presence of CPSP, physical function, mental function, return to function, pain severity, pain interference, and opioid medication use. Eligible patients were informed that their participation in the trial would last for 12 months and that if they were randomized to the CBT group, they could expect to spend approximately one hour per week for seven weeks completing the online modules. The COPE trial is registered at ClinicalTrials.gov (NCT03673358) and a more detailed description of COPE trial methods is available in our published protocol [4, 5]

## Participant enrollment

We used a convenience sample in which patients who declined to participate in the COPE trial were asked to consent to a brief study wherein they would complete a questionnaire that asked about previous experiences with research and mental health therapy, as well as their reasons for declining to participate. At the end of the questionnaire, they were offered the opportunity to opt-in to a telephone interview to further discuss their reasons for declining. The invitation to participate in this study was extended at the same time as the initial approach for the COPE trial and therefore took place either in person or by telephone. Verbal consent was obtained via telephone and recorded in the screening log, with both the individual obtaining consent and a witness signing a written documentation of the consent. A copy of the completed consent form was also offered to the participant. The recruitment period for this study began February 12th, 2021 and ended November 4th, 2022.

## Participation questionnaire

Participants who provided informed consent in person at the fracture clinic completed the questionnaire in clinic, either on paper or electronically using a secure electronic database (REDCap Cloud). Patients who consented via telephone were sent the questionnaire by secure email link and completed the questionnaire at home. The questionnaire took approximately five minutes to complete and covered the following topics: (1) demographics (age, gender, marital status, education, ethnicity, employment), (2) injury details, (3) previous experience with clinical research, (4) previous experience with CBT, (5) reasons for declining to participate in the COPE trial, (6) feelings of agreement with various statements regarding clinical research, CBT, and pain, and (7) concerns about future pain due to their injury. Finally, participants were asked if they were willing to be contacted by telephone for a brief interview to further discuss their reasons for declining to participate in the COPE trial.

## Qualitative interview methodology

A female research assistant with a Bachelor-level degree, (ML) experienced with the COPE trial and data collection in the fracture patient population conducted participant interviews by telephone. The interviewer had not had any contact with the participants before conducting the interview and introduced herself as a researcher associated with McMaster University. Prior to starting the interview, the interviewer ensured that participants were alone in a quiet space and had adequate time to complete the interview. Interviews followed a semi-structured guide that included the following topics: pain and recovery after a fracture, mental health therapy, research in general, and their experience being approached to participate in the COPE trial specifically. We aimed to interview a minimum of 20 patients, with interviews continuing until saturation; the point at which no new information was obtained from participants.

All interviews lasted approximately 20–30 minutes and, with the consent of participants, were digitally recorded. Recordings were transcribed verbatim, de-identified, and subsequently reviewed and verified by the interviewer. All participants were offered the opportunity

to read and revise their transcripts as a way of ensuring validity (member-checking). Five participants requested and were provided the transcripts from their interview, and none of those participants requested subsequent edits.

## Data analysis

We used a mixed methods approach in which the results from the qualitative phase of our study were used to help clarify and explain our quantitative findings. The questionnaire responses were analyzed using descriptive statistics. Categorical data were presented as counts (percentages), whereas continuous data were presented as means (and standard deviations [SDs]) when normally distributed or medians (and interquartile ranges) when not.

For the interviews, two research coordinators (JG and NF), who had not interviewed the participants, independently conducted data analysis in duplicate using Clarke and Braun's qualitative thematic analysis approach [7]. Thematic analysis is employed to identify, analyse, and report patterns. In this case, both JG and NF read all interviews multiple times, identified codes and identified tentative themes before merging the proposed themes. In instances where themes from each researcher were in conflict, we sought clarification from the interviewer and used a consensus-based decision making process. Recognizing that qualitative thematic analysis necessarily involves subjective interpretation on the part of the analysts, we strived to achieve descriptive and interpretive validity through the parallel analysis process, such that our findings reflected what the interviewees said [7, 8].

The interview transcripts were uploaded to Dedoose Version 9.0.17, a web application for managing, analyzing, and presenting qualitative and mixed method research data. Two research coordinators (JG and NF) independently read each of the interview transcripts and created a unique data codebook based on early impressions of the dataset. Next, the research coordinators compared and reconciled their results into one preliminary coding scheme and determined major themes and subthemes based on these codes. Where interview text was ambiguous, the audio files were consulted for further context prior to finalizing all themes and corresponding quotations. We then drew upon our qualitative and quantitative results jointly to come to a set of conclusions (i.e., 'meta-inferences') [9].

## Results

### Research participation questionnaire

Sixty-four of the approximately 360 patients (18%) who declined to participate in the COPE trial agreed to complete our questionnaire. The mean age of participants was 42 years (SD 16). Most participants were male (64%) and white (70%), and 31% had completed formal education to high school or less. Approximately three out of four participants (73%) were employed before their fracture. Lower limb fractures were the predominant injury (73%), with falls being the most common cause of injury (70%), followed by motor vehicle collisions (25%) (Table 1).

### Primary reasons for declining to participate in the cope trial

Almost half of participants reported the main reason they decided against participation in the COPE trial was the required time commitment (41%). Another 19% were disinterested in participating in clinical research, and 13% felt that they were unlikely to benefit from CBT. Less common reasons were reporting already receiving psychotherapy or support for their recovery (9%), and discomfort participating in CBT (6%). Only two participants (3%) felt uncomfortable with online technology, and one did not perceive that chronic pain after fracture was an issue. (Table 2).

**Table 1. Participant demographics and injury characteristics.**

|  | Total N = 64 participants |
|---|---|
| Age, years; mean (SD) | 42 (16) |
| Gender, n (%) |  |
| Male | 41 (64%) |
| Female | 23 (36%) |
| Marital status, n (%) |  |
| Married | 26 (41%) |
| Single | 18 (28%) |
| Living as common-law | 10 (16%) |
| Divorced | 4 (6%) |
| Widowed | 4 (6%) |
| Separated | 2 (3%) |
| Level of education, n (%) |  |
| 8th grade or less | 0 (0%) |
| 9th to 12th grade, no diploma | 7 (11%) |
| General education diploma or high school graduate | 13 (20%) |
| Some university/college, no degree | 16 (25%) |
| Associates degree (2-year degree) | 2 (3%) |
| Bachelors/college degree | 14 (22%) |
| Some graduate work, no degree | 1 (2%) |
| Graduate degree | 3 (5%) |
| Professional degree | 4 (6%) |
| Prefer not to answer | 4 (6%) |
| Ethnicity, n (%) |  |
| White | 45 (70%) |
| South Asian | 7 (11%) |
| Prefer not to answer | 6 (9%) |
| Middle Eastern | 3 (5%) |
| Black | 1 (2%) |
| Latino | 1 (2%) |
| Multiracial | 1 (2%) |
| Employment, n (%) |  |
| Yes | 47 (73%) |
| No | 17 (27%) |
| Location, n (%)* |  |
| Upper Body (i.e. shoulder, arm, hand) | 22 (34%) |
| Lower Body (i.e. pelvis, leg, ankle, foot) | 47 (73%) |
| Cause of injury, n (%) |  |
| Motor vehicle collision | 16 (25%) |
| Fall | 45 (70%) |
| Sports injury | 3 (5%) |

*More than one response can be selected

## Participant experience with clinical trials and CBT

When asked about previous experience with clinical trials, one-third of the participants (34%) indicated that they had been invited to participate in a prior clinical research study, and 77% of those had agreed to take part in the study. Only 6 participants (9%) had previously received

**Table 2. Reasons for declining to participate in the COPE trial.**

| | Total N = 64 participants |
|---|---|
| Participating in the study requires too large a time commitment, n (%) | 31 (48%) |
| I do not wish to take part in clinical research, n (%) | 12 (19%) |
| I do not feel that I will benefit from cognitive behavioural therapy, n (%) | 8 (13%) |
| Already receiving psychotherapy or support from outside sources, n (%) | 6 (9%) |
| I do not feel comfortable participating in cognitive behavioural therapy, n (%) | 4 (6%) |
| I do not feel familiar/comfortable enough with technology to register and complete the cognitive behavioural therapy program online, n (%) | 2 (3%) |
| I do not think that chronic pain is a real problem after a fracture, n (%) | 1 (2%) |

CBT, for anxiety (n = 3), depression (n = 1), posttraumatic stress disorder (n = 1), or substance use disorder (n = 1) (Table 3).

## Participants attitudes and beliefs regarding clinical trials and CBT

Almost half of the participants (47%) were disinterested in participating in clinical research. Although most participants stated they were familiar enough with technology to be able to register in an online CBT program (61%) and 47% of participants agreed that they felt comfortable with participating in CBT, 64% agreed that they did not have enough time to participate in the trial. Participants were divided as to whether they felt they would benefit from CBT: 30% believed they would, 36% were uncertain, and 34% felt they would not. Additionally, 39% were unwilling to be randomly assigned to treatment or control groups.

Most participants did not have concerns about the privacy or confidentiality of CBT (75%). Ninety-two percent of participants indicated that they liked the way they were approached by research personnel, and 80% were happy with their clinical care. When asked about their beliefs regarding chronic pain after a fracture, 63% of respondents agreed that it could be a real problem after a fracture, while 31% answered they were uncertain. Only 6% of participants expressed the belief that chronic pain could not be a real issue after a fracture. Regarding

**Table 3. Participant experience with clinical trials and cognitive behavioural therapy.**

| | Total N = 64 participants |
|---|---|
| Previously invited to participate in clinical research, n (%) | |
| Yes | 22 (34%) |
| Agreed to participate | 17 (77%) |
| Declined to participate | 5 (23%) |
| Too busy to participate | 2 (40%) |
| Not interested | 3 (60%) |
| No | 42 (66%) |
| Previously participated in cognitive behavioural therapy, n (%) | |
| Yes, for | 6 (9%) |
| Anxiety | 3 (50%) |
| Depression | 1 (17%) |
| Posttraumatic Stress Disorder | 1 (17%) |
| Substance Use Disorders | 1 (17%) |
| No | 58 (91%) |

participants' concerns about developing chronic pain after their fracture, 45% were concerned, 30% were uncertain, and 25% were unconcerned (Table 4).

## Qualitative interviews

The first 34 participants who completed the questionnaire were given the option to participate in an interview, of which 20 agreed (59% response rate). The mean age of interview participants was 43 years (SD 13 years) and the majority were male (75%).

There were four main themes that emerged from the interviews with participants: 1) the absence of the need for mental health support and the ability to cope with recovery on one's own; 2) the belief that CBT might be appropriate for others, but not for oneself; 3) preference for in-person or online delivery of CBT; and 4) concerns with the study's required time commitment (Figs 1 and 2).

**Theme 1—Absence of a need for mental health support and ability to cope on one's own.** The first identified theme was the absence of the need for mental health support and the ability to cope with recovery on one's own. Eight participants interviewed by research personnel had similar mindsets believing they could overcome challenges without external help. Some participant testimonies surrounding this theme included:

*"I got a strong mind. I can probably get through on my own."*

*(Male 20 years)*

*"I can deal with it and work things out. I am not worried about it."*

*(Male 48 years)*

*"I can deal with these problems myself and I don't need anyone else's help to deal with my emotional problems."*

*(Male 53 years)*

One participant described the role that culture and upbringing played in their beliefs regarding their need for mental health support:

*"But like some people, like due to parents, how they were brought up, they were instilled different mentalities, you know, to put in their minds that like these things don't really exist. . .. I believe that because of that same upbringing, is the reason why I think I don't need it."*

*(Male 35 years)*

**Theme 2—Cognitive behavioural therapy may be appropriate for others.** Regarding the theme of thinking that CBT might be appropriate for others, but not for oneself, 11 interviewees expressed this with various statements including:

*"It will be helpful for lots of people, depends on their mental state, we are all different."*

*(Female 55 years)*

*"For me, no. But I could see other people, yes."*

*(Male 53 years)*

**Table 4. Participants attitudes and beliefs regarding clinical trials and cognitive behavioural therapy.**

| | Total N = 64 participants |
|---|---|
| I do not wish to take part in clinical research, n (%) | |
| Strongly Disagree | 8 (13%) |
| Disagree | 14 (22%) |
| Uncertain | 12 (19%) |
| Agree | 24 (38%) |
| Strongly Agree | 6 (9%) |
| I do not feel comfortable participating in cognitive behavioural therapy, n (%) | |
| Strongly Disagree | 13 (20%) |
| Disagree | 17 (27%) |
| Uncertain | 14 (22%) |
| Agree | 13 (20%) |
| Strongly Agree | 7 (11%) |
| Participating in the study requires too large a time commitment, n (%) | |
| Strongly Disagree | 2 (3%) |
| Disagree | 9 (14%) |
| Uncertain | 12 (19%) |
| Agree | 20 (31%) |
| Strongly Agree | 21 (33%) |
| I do not feel familiar/comfortable enough with technology to register and complete the cognitive behavioural therapy program online, n (%) | |
| Strongly Disagree | 21 (33%) |
| Disagree | 18 (28%) |
| Uncertain | 8 (13%) |
| Agree | 9 (14%) |
| Strongly Agree | 8 (13%) |
| I do not feel that I will benefit from cognitive behavioural therapy, n (%) | |
| Strongly Disagree | 7 (11%) |
| Disagree | 15 (23%) |
| Uncertain | 23 (36%) |
| Agree | 14 (22%) |
| Strongly Agree | 5 (8%) |
| I do not wish to be randomly assigned to one of the two treatment groups (usual care or cognitive behavioural therapy), n (%) | |
| Strongly Disagree | 8 (13%) |
| Disagree | 17 (27%) |
| Uncertain | 14 (22%) |
| Agree | 19 (30%) |
| Strongly Agree | 6 (9%) |
| The cognitive behavioural therapy is being delivered through online modules (as compared to in person sessions), n (%) | |
| Strongly Disagree | 7 (11%) |
| Disagree | 15 (23%) |
| Uncertain | 27 (42%) |
| Agree | 11 (17%) |
| Strongly Agree | 4 (6%) |
| I have concerns regarding privacy and/or confidentiality, n (%) | |
| Strongly Disagree | 20 (31%) |

(*Continued*)

**Table 4.** (Continued)

| | Total N = 64 participants |
|---|---|
| Disagree | 28 (44%) |
| Uncertain | 8 (13%) |
| Agree | 5 (8%) |
| Strongly Agree | 3 (5%) |
| I did not like the way that I was approached by research personnel, n (%) | |
| Strongly Disagree | 41 (64%) |
| Disagree | 18 (28%) |
| Uncertain | 4 (6%) |
| Agree | 0 (0%) |
| Strongly Agree | 1 (2%) |
| I am unhappy with my clinical care, n (%) | |
| Strongly Disagree | 31 (48%) |
| Disagree | 20 (31%) |
| Uncertain | 7 (11) |
| Agree | 5 (8%) |
| Strongly Agree | 1 (2%) |
| I do not think that chronic pain is a real problem after a fracture, n (%) | |
| Strongly Disagree | 19 (30%) |
| Disagree | 21 (33%) |
| Uncertain | 20 (31%) |
| Agree | 3 (5%) |
| Strongly Agree | 1 (2%) |
| Concerned about continuing to have pain, n (%) | |
| Not concerned at all | 5 (8%) |
| Slightly concerned | 11 (17%) |
| Neutral | 19 (30%) |
| Somewhat concerned | 18 (28%) |
| Very concerned | 11 (17%) |

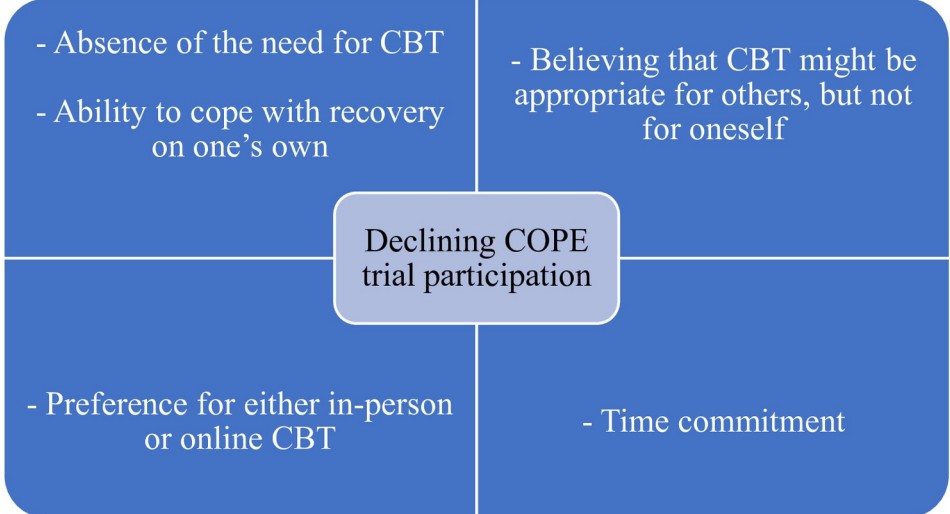

**Fig 1. Thematic map of themes identified from individual interviews.**

| | | |
|---|---|---|
| **MENTAL HEALTH** | **I don't need therapy / I can deal with it myself** | • I have other coping mechanisms (1)<br>• **"Old-school" mentality / I don't need help (26)**<br>• **Others could benefit from therapy, not me (27)** |
| | Mental health and pain (or physical health) are related | • General description of feelings about pain (21)<br>• Positive belief that mental health treatment could also help with physical pain (5) |
| | Positive views of therapy / mental health treatment | • Miscellaneous positive statements about therapy (13)<br>• There is strength in seeking help (2)<br>• Therapy works if you want to do it (1) |
| | Negative views of therapy / mental health treatment | • Miscellaneous negative statements about therapy (5)<br>• I have experience with a bad / mismatched therapist (3)<br>• Therapy requires drugs, and I'm against that (2)<br>• Lack of understanding about mental health therapy / CBT intervention (6) |
| | **Thoughts about online modality** | • **In-person is better for me (10)**<br>• **There is no accountability in an online therapy program (2)**<br>• **Neutral or positive statements about online modality (7)** |
| **RESEARCH** | **Reasons for declining the COPE study** | • **Time commitment (24)**<br>• The screening and consent process needs improvement (6)<br>• I didn't understand the trial (1)<br>• Concerns about randomizing to usual care (1) |
| | Reasons participants considered in favour of participating | • Obligation (5)<br>• Contributing to medical science specifically (5)<br>• Helping others more generally (10)<br>• Benefits me personally (2) |
| | Thoughts about reimbursement | • Reimbursements are motivating (5)<br>• Reimbursements don't matter to me one way or another (15) |

**Fig 2. Coding tree.**

Some participants detailed specifically how they differed from the kind of person who might benefit from CBT. These reasons included an "old school" mentality and strong existing social support systems:

*"You know I'm old school, so it's just, you know, suck it up and deal with it. But I know a lot of people can't deal with that." (Male 53 years)*

*"I imagine it to be very helpful to people who are alone."*

*(Female 71 years)*

*"It's good if you don't have someone to talk to about it, like to deal with your daily struggles and everything. So, I guess it is very beneficial for an individual who doesn't have anybody."*

*(Male 24 years)*

Finally, people with mental illness were perceived as potential beneficiaries of CBT:

*"If they have a weaker mentality and they are more susceptible to anxiety, it will have a great impact."*

*(Male 30 years)*

*"I guess people with depression would benefit the most from this therapy."*

*(Female 55 years)*

**Theme 3—Preferences for online or in-person CBT.**   Participant opinions were divided regarding online CBT. Five participants believed having an online CBT option was favourable compared to in-person CBT, in terms of reducing the time commitment and removing the need to travel to an appointment. Some participant testimonies regarding this theme are as follows:

*"Beneficial to finding the time to do it online."*

*(Female 65 years)*

*"Face-to-face meeting requires a specific time slot, it's not as easy to do"*

*(Male 31 years)*

However, 11 interviewees indicated that meeting with a therapist in person was favourable as they believed it would help improve communication and be better received by participants. Additionally, three people referred to lack of discipline as one of the reasons for supporting the in-person option, as it required greater accountability.

*"Dealing with mental issues, I think it's always better to talk to somebody. The internet is very impersonal."*

*(Male 51 years)*

*"I'm not disciplined enough to do it myself."*

*(Female 53 years).*

**Theme 4—Time commitment.**   Most participants (60%) were concerned about the amount of time the CBT and trial would require.

*"The COPE study was a pretty long commitment. . .I'm always busy with work and with the kids."*

*(Male 48 years)*

*"That's a long period of time commitment."*

*(Female 55 years)*

One participant also identified that during a period of recovery from trauma, time commitments in general can seem especially taxing:

*"Given the people that are being approached for the study are already in pain or dealing with some sort of trauma. It's just another thing to do. Feels overwhelming at times."*

*(Male 32 years)*

## Discussion

Sixty-four patients who declined to participate in the COPE trial participated in the current study. Excessive time commitment (48%) was the main reason for not participating in the COPE trial, followed by disinterest in participating in clinical research (19%). Four main themes emerged from interviews with participants: 1) the absence of the need for additional mental health support and the ability to cope with problems on one's own; 2) the belief that CBT might be appropriate for others, but not for oneself; 3) preference for either in-person or online CBT; and 4) concerns with the trial's required time commitment.

### Comparison with existing literature

Many interviewees expressed the belief that therapy is beneficial for others, but not for themselves or that only people with a diagnosed mental illness need psychotherapy. This language reflects a lower level of mental health literacy, which affects willingness to seek care and belief that such care will be effective [10]. A 2021 review of behavioral treatments for migraine headache noted that a low level of knowledge and acceptance of CBT as a treatment for non-psychological diagnosis reduced utilization by both physicians and patients [11]. The desire of participants to distance themselves from others, often implied as having severe symptoms, who may need therapy is mirrored in Hantzi et. al's 2019 study which found that self-stigma and essentialist beliefs about mental health have a direct link to help-seeking behaviours [12]. Such attitudes may result in under-utilization of CBT to manage pain. These findings suggest strategies to address self-stigma and increase help-seeking behaviours regarding mental health may be helpful to encourage participation in trials of CBT. Strategies that have proven the most effective at reducing mental health stigma, such as public awareness campaigns and robust legal protections, often produce slow change and are beyond the scope of individual healthcare institutions [13]. A smaller-scale option that shows potential is peer-led interventions [14]. For example, having a peer support group to complement the CBT intervention could reduce self-stigma in participants.

Another theme that emerged from participants was the time commitment of CBT. Time commitment has been cited as one of the main reasons for declining to participate in health research [15, 16]. Suggested strategies include enrolling more participants to account for withdrawal, providing a simple timeline of study commitments, offering flexible starting times, and re-contacting potential participants at a later date [15, 16]. Additionally, future researchers might consider adapting a CBT program to deliver the core CBT tenets in a brief, but efficacious way that would be acceptable to fracture patients.

Lastly, some participants had preferences regarding the delivery method of CBT, with some preferring in-person sessions and others internet-based therapy; however, the pilot phase of the COPE trial established that conducting in-person CBT was not feasible for the fracture patient population [17]. For that reason, the decision was made to switch to an online CBT provider for the definitive phase of the COPE trial. Some advantages cited by patients that receive internet-delivered CBT included more control over the time, pace, and location of the therapy [18]. Some patients will prefer in-person CBT, and the ability to offer either format to potential study participants may improve recruitment. Additionally, a recent systematic review

of randomized controlled trials found high certainty evidence for no difference in effectiveness between in-person CBT or remote delivery CBT with therapist guidance, across a range of clinical conditions [19, 20].

## Study limitations

This study took place at one site during the first year of the multi-year, multi-site COPE trial and therefore was offered to a minority of all patients who declined to participate. Thus, the sample included may not be representative of all decliners to the COPE trial. This study required significantly less time than the COPE trial and may have been particularly appealing to those fracture patients that were interested in research but truly did not have time in their schedules to commit to a larger trial. Finally, there is an interesting contradiction in the views of the participants of this study as they represent a group of people who technically did agree to participate in research, speaking about why they declined to participate in research. It is worth remembering that there remains a group of 'true decliners' that opted not to participate in either the COPE trial or the present study.

## Implications for future studies

Our findings have the potential to help future clinical trials optimize recruitment when exploring the effectiveness of mental health interventions for trauma patients. Firstly, giving participants examples of how CBT is helpful for problems unrelated to mental health may help reduce stigma associated with psychotherapy. Implementing the time commitment strategies from Levickis et al., such as providing a simple and transparent schematic of study commitments (e.g. timeline), could also encourage potential participants to join the study [16]. Researchers may also want to consider providing options for both in-person and online CBT to reduce selection bias.

## Conclusions

To maximize enrollment, orthopaedic trials involving CBT should optimize participants' time commitment and provide the option of participating in-person or online. Providing clarity about the demands and content of the CBT for pain management program during the consent process may also prove helpful in promoting enrollment.

## Acknowledgments

*The COPE Investigators

**Steering Committee:** Jason W. Busse (Principal Investigator, McMaster University, Hamilton, ON), Sheila Sprague (Principal Investigator, McMaster University, Hamilton, ON), Mohit Bhandari (McMaster University, Hamilton, ON), Gerard Slobogean (University of Maryland School of Medicine, Baltimore, MD), Lehana Thabane (McMaster University, Hamilton, ON), Randi E. McCabe (McMaster University, Hamilton, ON), Emil H. Schemitsch (University of Western Ontario, London, ON).

**Research Methodology Advisory Core:** Gordon H. Guyatt (McMaster University, Hamilton, ON), PJ Devereaux (McMaster University, Hamilton, ON), Lehana Thabane (McMaster University, Hamilton, ON), Mohit Bhandari (McMaster University, Hamilton, ON).

**Orthopaedic Surgery Advisory Core:** I. Leah Gitajn (Dartmouth University, Hanover, NH), Gerard Slobogean (University of Maryland School of Medicine, Baltimore, MD), Emil H. Schemitsch (University of Western Ontario, London, ON).

**Psychology Advisory Core:** Randi E. McCabe (McMaster University, Hamilton, ON), Matilda Nowakowski (St. Joseph's Healthcare, Hamilton, ON), Eleni Hapidou (McMaster University, Hamilton, ON), Delia Chiaramonte (Greater Baltimore Medical Center, Baltimore, MD).

**Medica/Pain Advisory Core:** Henrick Kehlet (Copenhagen University, Denmark), James Khan (Stanford University, Stanford, CA).

**Adjudication Committee:** Matilda Nowakowski (St. Joseph's Healthcare, Hamilton, ON), Aresh Sepehri (University of Maryland, Baltimore, MD).

**Methods Centre, McMaster University, Hamilton, ON:** Paula McKay, Gina Del Fabbro, Natalie Fleming, Christy Shibu, Brittney Kay, Diane Heels-Ansdell, Sofia Bzovsky.

**Hamilton Health Sciences—General Site, Hamilton, ON:** Brad A. Petrisor, Dale Williams, Bill Ristevski, Jamal Al-Asiri, Herman Johal, Matthew Denkers, Kris Rajaratnam, Jodi L. Gallant, Sarah MacRae, Kaitlyn Pusztai, Sara Renaud, Nicki Johal.

**The Ottawa Hospital, Ottawa, ON:** Steven Papp, Karl-Andre Lalonde, Bradley Meulenkamp, Allan Liew, Manisha Mistry, Braden Gammon, Wade Gofton, Geoffrey Wilkin, Melanie Dodd-Moher.

**Thunder Bay Regional Health Sciences Centre, Thunder Bay, ON:** David Puskas, Travis Marion, Tina Lefrancois, Jubin Payandeh, Claude Cullinan, Tracy Wilson, Kurt Droll, Michael Riediger, Rabail Siddiqui, Shalyn Littlefield, Simrun Chahal, Paige Wagar.

**Foothills Medical Centre, University of Calgary, Calgary, AB:** Prism S. Schneider, Tosin Ogunleye, Tanya Cherppukaran, Karin Lienhard.

**Memorial University, St. John's, NFLD:** Nicholas Smith, Sarah Anthony, Krista Butt.

**University of Maryland, R Adams Cowley Shock Trauma Center, Baltimore, MD:** Gerard Slobogean, LaShann Selby, Murali Kovvur, Joshua Lawrence, Skyler Sampson, Kristin Turner, Alice Bell, Vivian Li and David Okhuereigbe

**University of Maryland Capital Region Health, Baltimore, MD:** Todd Jaeblon, Haley K. Demyanovich, Sneh Talwar.

**Dartmouth-Hitchcock Medical Center, Lebanon, NH:** I. Leah Gitajn, Devin Mullin, Holly Symonds, Jon Mikael Anderson, Daniel Austin, Logan Bateman, Gerard Chang, Marcus Coe, Frances Faro, Alexander Orem, Philip Wolinsky

**Beth Israel Deaconess Medical Center, Boston, MA:** Paul J. Appleton, John J. Wixted, Edward K. Rodriguez, Michael F. McTague, Katiri Wagner, Kristina Brackpool, Kate Hegermiller, Nhi Nguyen.

**Indiana University School of Medicine, Indianapolis, IN:** Roman M. Natoli, MaKenzie G. Barger, Jena L. Robinson.

**Prisma Health–Upstate, Greenville, SC:** Kyle J. Jeray, Thomas M. Schaller, Michael S. Sridhar, John D. Adams, Richard W. Gurich Jr., Michelle Drosback, Emily Williams, Sabrina Salley, Calleigh Brignull, Harper Sprouse, Rosalynn Rice, Kyle Adams

**St. Joseph's Hospital, Hamilton, ON:** Giuseppe Valente, James Yan, Kim Madden, Kim Irish, Nathasha Rajapaksege, Yusra Aslam, Melanie Mark, Avneet Dhiman

## Author Contributions

**Conceptualization:** Sheila Sprague, Jodi L. Gallant, Sarah MacRae, Paula McKay.

**Data curation:** Sheila Sprague, Sofia Bzovsky, Mavis Lyons, Jason W. Busse.

**Formal analysis:** Sheila Sprague, Jodi L. Gallant, Natalie Fleming, Sofia Bzovsky, Sarah MacRae, Jason W. Busse.

**Funding acquisition:** Sheila Sprague, Paula McKay.

**Investigation:** Sheila Sprague, Jodi L. Gallant, Sarah MacRae, Mavis Lyons, Herman Johal.

**Methodology:** Sheila Sprague, Jodi L. Gallant, Sofia Bzovsky, Sarah MacRae, Paula McKay, Jason W. Busse.

**Project administration:** Sheila Sprague, Jodi L. Gallant, Sarah MacRae.

**Resources:** Sheila Sprague, Herman Johal, Paula McKay, Jason W. Busse.

**Supervision:** Sheila Sprague, Herman Johal, Paula McKay, Jason W. Busse.

**Validation:** Sheila Sprague.

**Writing – original draft:** Sheila Sprague, Jodi L. Gallant, Natalie Fleming, Sofia Bzovsky, Jose Manuel De Maria Prieto.

**Writing – review & editing:** Sheila Sprague, Jodi L. Gallant, Natalie Fleming, Sofia Bzovsky, Jose Manuel De Maria Prieto, Herman Johal, Paula McKay, Jason W. Busse.

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
