## [Decision Letter · Decision Letter 0]

28 Aug 2024

PONE-D-24-19470Reasons for declining to participate in a trial of online cognitive behavioural therapy following orthopaedic trauma: A mixed methods studyPLOS ONE

Dear Dr. Sprague,

Thank you for submitting your manuscript to PLOS ONE. After careful consideration, we feel that it has merit but does not fully meet PLOS ONE’s publication criteria as it currently stands. Therefore, we invite you to submit a revised version of the manuscript that addresses the points raised during the review process.

We look forward to receiving your revised manuscript.

Kind regards,

Runtang Meng, PhD

Academic Editor

PLOS ONE

3. One of the noted authors is a group or consortium [COPE Investigators]. In addition to naming the author group, please list the individual authors and affiliations within this group in the acknowledgments section of your manuscript. Please also indicate clearly a lead author for this group along with a contact email address.

Reviewers' comments:

Reviewer's Responses to Questions

**Comments to the Author**

1. Is the manuscript technically sound, and do the data support the conclusions?

Reviewer #1: Yes

Reviewer #2: Yes

2. Has the statistical analysis been performed appropriately and rigorously? 

Reviewer #1: Yes

Reviewer #2: Yes

3. Have the authors made all data underlying the findings in their manuscript fully available?

Reviewer #1: Yes

Reviewer #2: Yes

4. Is the manuscript presented in an intelligible fashion and written in standard English?

Reviewer #1: Yes

Reviewer #2: Yes

5. Review Comments to the Author

Reviewer #1: This manuscript presents a well-conducted mixed-methods study examining reasons why patients decline to participate in a trial of online cognitive behavioral therapy (CBT) following orthopedic trauma. The study design, methods, and analysis are sound, and the findings provide valuable insights that could help improve recruitment for future trials in this area.

Here are some suggestions:

1.Consider providing more details on the COPE trial intervention in the methods section to give readers better context.

2.The discussion could be strengthened by further exploring potential strategies to address the identified barriers to participation, particularly around mental health stigma and time commitment concerns.

Overall, this is a well-executed study that makes an important contribution to understanding barriers to participation in CBT trials for orthopedic trauma patients. The findings have clear implications for improving recruitment in future studies. I recommend publication with minor revisions to address the suggestions above.

Reviewer #2: Sheila Sprague et al. explored the reasons for refusal of the COPE trial and population characteristics of fracture patients through questionnaires and telephone calls to provide information about evidence of psychotherapy and orthopaedic trauma recovery for the subsequent patient consent process for the COPE trial to facilitate patient registration. There were no obvious errors in the writing of the study and the language used was fair, but the readability of the study was more general.

6. PLOS authors have the option to publish the peer review history of their article (what does this mean?). If published, this will include your full peer review and any attached files.

Reviewer #1: **Yes: **Zongyou Yang

Reviewer #2: No

---

## [Author Response · Author response to Decision Letter 0]

18 Sep 2024

September 3, 2024 

RE: Reasons for Declining to Participate in a Trial of Online Cognitive Behavioural Therapy Following Orthopaedic Trauma: A Mixed Methods Study 

Dear Editor, 

Thank you for considering our manuscript entitled, “Reasons for Declining to Participate in a Trial of Online Cognitive Behavioural Therapy Following Orthopaedic Trauma: A Mixed Methods Study” for publication in PLOS ONE. 

Please find attached both clean and tracked versions of our revised manuscript. We have responded to the journal and reviewer comments below. 

Response: We have reviewed our file names and formatting and have confirmed that they are in line with PLOS ONE’s requirements. Please specify if anything has been missed. 

We note that you have indicated that there are restrictions to data sharing for this study. PLOS only allows data to be available upon request if there are legal or ethical restrictions on sharing data publicly. 

Response: Our ethics board and institutional policies require data to be kept confidential unless a request for data sharing is submitted and approved. Anyone interested in viewing the study data can send a request to corresponding author via email. 

One of the noted authors is a group or consortium [COPE Investigators]. In addition to naming the author group, please list the individual authors and affiliations within this group in the acknowledgments section of your manuscript. Please also indicate clearly a lead author for this group along with a contact email address. 

Response: Please find the individual authors and affiliations of the COPE Investigators group listed under Acknowledgments. Lead authors are Dr. Sheila Sprague (sprags@mcmaster.ca) and Dr. Jason Busse (bussejw@mcmaster.ca). This information is also provided on the title page. 

Consider providing more details on the COPE trial intervention in the methods section to give readers better context. 

Response: We have added more details on the CBT program as requested. 

The discussion could be strengthened by further exploring potential strategies to address the identified barriers to participation, particularly around mental health stigma and time commitment concerns. 

Response: We have expanded our discussion on these topics as requested. 

Please do not hesitate to contact me if you have any questions or concerns. Thank you for your consideration and we look forward to further communication with you. 

Sincerely, 

Sheila Sprague, PhD Jason W. Busse, DC, PhD 

293 Wellington St. N, Suite 110 1280 Main St. West, HSC-2V9 

Hamilton, ON L8L 8E7 Hamilton, ON L8S 4K1 

Email: sprags@mcmaster.ca Email: bussejw@mcmaster.ca

Tel: 905-929-7846 Tel: 905-525-9140 (x21731)

---

## [Decision Letter · Decision Letter 1]

20 Nov 2024

PONE-D-24-19470R1Reasons for declining to participate in a trial of online cognitive behavioural therapy following orthopaedic trauma: A mixed methods studyPLOS ONE

Dear Dr. Sprague,

Thank you for submitting your manuscript to PLOS ONE. After careful consideration, we feel that it has merit but does not fully meet PLOS ONE’s publication criteria as it currently stands. Therefore, we invite you to submit a revised version of the manuscript that addresses the points raised during the review process.

We look forward to receiving your revised manuscript.

Kind regards,

Runtang Meng, PhD

Academic Editor

PLOS ONE

Journal Requirements:

Reviewers' comments:

Reviewer's Responses to Questions

**Comments to the Author**

1. If the authors have adequately addressed your comments raised in a previous round of review and you feel that this manuscript is now acceptable for publication, you may indicate that here to bypass the “Comments to the Author” section, enter your conflict of interest statement in the “Confidential to Editor” section, and submit your "Accept" recommendation.

Reviewer #1: All comments have been addressed

Reviewer #3: (No Response)

2. Is the manuscript technically sound, and do the data support the conclusions?

Reviewer #1: Yes

Reviewer #3: Yes

3. Has the statistical analysis been performed appropriately and rigorously? 

Reviewer #1: N/A

Reviewer #3: Yes

4. Have the authors made all data underlying the findings in their manuscript fully available?

Reviewer #1: Yes

Reviewer #3: No

5. Is the manuscript presented in an intelligible fashion and written in standard English?

Reviewer #1: Yes

Reviewer #3: Yes

6. Review Comments to the Author

Reviewer #1: The authors have done a good job addressing the comments. They expanded on the COPE trial intervention details, enhancing the reader’s understanding of the study’s methodology, and added valuable insights into the barriers to participation. The manuscript is technically sound, with data that supports the conclusions. The manuscript is clearly written. Overall, the revisions have made the manuscript more compelling, and I believe it’s now suitable for publication.

Reviewer #3: The authors addressed the following two comments from the first reviewer satisfactorily.

1) Consider providing more details on the COPE trial intervention in the methods section

to give readers better context.

2) The discussion could be strengthened by further exploring potential strategies to

address the identified barriers to participation, particularly around mental health stigma

and time commitment concerns.

Additional comments

Line 79-81, report "This study was conducted at one clinical site currently participating in the COPE trial (Hamilton Health Sciences – General Site) and followed the Consolidated criteria for reporting qualitative research (COREQ) criteria (5).

The following comments relate to the methods section. Additional clarity in reporting methods would strengthen the reporting of methods.

Line 92, recommends changing "will complete" to complete in past tense.

Line 149-151: Did any participants review their transcriptions? If so, did they agree that responses were accurately captured?

Line 161-162: Can you describe more specifically how you utilized Braun and Clarke's thematic analysis for your qualitative analysis?

Line 165-166: How did you strive to achieve descriptive and interpretive validity?

Line 169-171: Can you describe the coding process in more detail?

Questions to consider relating to COREQ reporting requirements include the following:

Domain 1: Research team and reflexivity

Interviewer characteristics: Bias, assumptions etc.

Domain 3: analysis and findings

Consider including a description of the coding tree.

Consider describing in more detail how applied Braun and Clarke's methods for thematic analysis.

Was the semi-structured interview guide piloted? If yes, with whom?

Thank you for this opportunity to review this interesting study.

7. PLOS authors have the option to publish the peer review history of their article (what does this mean?). If published, this will include your full peer review and any attached files.

Reviewer #1: **Yes: **Zongyou Yang

Reviewer #3: No

---

## [Author Response · Author response to Decision Letter 1]

4 Dec 2024

1. Please review your reference list to ensure that it is complete and correct.

Response: We have reviewed the reference list and added corrections for two papers (references 4 and 19) that have been published recently. None of the corrections change the utility of the cited research, so we have simply added each correction alongside the original reference.

2. Line 92, recommend changing "will complete" to complete in past tense.

Response: We have made this change accordingly at line 92.

3. Line 149-151: Did any participants review their transcriptions? If so, did they agree that responses were accurately captured?

Response: The option to read and revise the transcribed interview was offered to all participants. Five participants opted to receive copies of their interview transcriptions to review. None of the five requested changes to the transcript. This information has been clarified at lines 149-152.

4. Line 161-162: Can you describe more specifically how you utilized Braun and Clarke's thematic analysis for your qualitative analysis? / Consider describing in more detail how applied Braun and Clarke's methods for thematic analysis.

Response: Thank you for this revision – we have added more detail about the theming process at lines 162-164.

5. Line 165-166: How did you strive to achieve descriptive and interpretive validity?

Response: Thank you for this question. We have added additional detail about the separate and parallel theming process that provided increased validity to our results at line 168-169.

6. Line 169-171: Can you describe the coding process in more detail?

Response: Thank you for this question. Through responding to the previous 2 questions, we have added more detail about the coding process in lines 161-169.

7. Consider including the following (for COREQ Domain 1: Research team and reflexivity): Interviewer characteristics: Bias, assumptions etc.

Response: Thank you for this question. We have added some information about the education level of the interviewer to lines 136-140. This now represents all the information we collected about the interviewer at the time of interview.

8. Consider including a description of the coding tree.

Response: We have created a coding tree and added it as Figure 2, attached with this resubmission.

9. Was the semi-structured interview guide piloted? If yes, with whom?

Response: Thank you for this question. The interview was piloted with members of the research team who regularly work with fracture patients.

---

## [Decision Letter · Decision Letter 2]

31 Dec 2024

Reasons for declining to participate in a trial of online cognitive behavioural therapy following orthopaedic trauma: A mixed methods study

PONE-D-24-19470R2

Dear Dr. Sprague,

We’re pleased to inform you that your manuscript has been judged scientifically suitable for publication and will be formally accepted for publication once it meets all outstanding technical requirements.

Kind regards,

Runtang Meng, PhD

Academic Editor

PLOS ONE

Additional Editor Comments (optional):

Reviewers' comments:

Reviewer's Responses to Questions

**Comments to the Author**

1. If the authors have adequately addressed your comments raised in a previous round of review and you feel that this manuscript is now acceptable for publication, you may indicate that here to bypass the “Comments to the Author” section, enter your conflict of interest statement in the “Confidential to Editor” section, and submit your "Accept" recommendation.

Reviewer #1: All comments have been addressed

Reviewer #3: All comments have been addressed

2. Is the manuscript technically sound, and do the data support the conclusions?

Reviewer #1: Yes

Reviewer #3: (No Response)

3. Has the statistical analysis been performed appropriately and rigorously? 

Reviewer #1: N/A

Reviewer #3: (No Response)

4. Have the authors made all data underlying the findings in their manuscript fully available?

Reviewer #1: Yes

Reviewer #3: (No Response)

5. Is the manuscript presented in an intelligible fashion and written in standard English?

Reviewer #1: Yes

Reviewer #3: (No Response)

6. Review Comments to the Author

Reviewer #1: (No Response)

Reviewer #3: (No Response)

7. PLOS authors have the option to publish the peer review history of their article (what does this mean?). If published, this will include your full peer review and any attached files.

Reviewer #1: **Yes: **Zongyou Yang

Reviewer #3: No

---

## [Editor Report · Acceptance letter]

8 Jan 2025

PONE-D-24-19470R2 

PLOS ONE

Dear Dr. Sprague, 

I'm pleased to inform you that your manuscript has been deemed suitable for publication in PLOS ONE. Congratulations! Your manuscript is now being handed over to our production team.

Kind regards, 

on behalf of

Dr. Runtang Meng 

Academic Editor

PLOS ONE